Submission

# Non-Topological Edge-Localized Yu-Shiba-Rusinov States in CrBr$_3$/NbSe$_2$ Heterostructures

**Jan P. Cuperus, Daniel Vanmaekelbergh and Ingmar Swart$^\star$**

Debye Institute for Nanomaterials Science, Utrecht University, The Netherlands

$\star$ i.swart@uu.nl

## Abstract

Topological superconductivity is predicted to emerge in certain magnet-superconductor hybrid systems. Here, we revisit the heterostructure of insulating monolayer CrBr$_3$ and NbSe$_2$, for which different conclusions on the presence of topological superconductivity have been reported. Using low-temperature scanning tunneling microscopy and (shot noise) spectroscopy, we find that the superconducting gap well inside the CrBr$_3$ islands is not affected by magnetism. At the island edges, we observe Yu-Shiba-Rusinov (YSR) states at a variety of in-gap positions, including zero energy. The absence of topological superconductivity is verified by extensive dI/dV measurements at the CrBr$_3$ island edges. Our results ask for a more detailed understanding of the interaction between magnetic insulators and superconductors.

# 1 Introduction

At present, the road to scalable quantum computing is cumbersome due to the qubit decoherence problem [1, 2]. Despite recent advancements in quantum error correction, the decoherence problem requires large physical and technological overhead [3, 4]. The development of topological qubits, which have significantly lower decoherence rates, would therefore be a leap forward towards scalable quantum computing. Central to the working principle of topological qubits are Majorana zero-modes (MZMs), exotic fermionic particles that obey non-Abelian statistics [5, 6]. As a potential platform for the realization of MZMs, topological superconductors are actively sought after in condensed matter systems. Examples of material systems that have shown signs of MZMs include proximitized semiconductor nanowires [7–10], proximitized topological insulators [11–14], superconducting topological surface states [15, 16], and 1D/2D magnetic lattices on superconducting surfaces [17–23]. In the examples above, topological superconductivity (TSC) is established by the induced, effective p-wave pairing of superconductivity; the required spin degeneracy lifting is provided by a topological phase transition [5], as has been shown for lattices of YSR states [23]. YSR states are in-gap bound states induced by the exchange interaction between magnetic atoms (historically: impurities) and Cooper pairs [24–27]. In the classical spin approximation [28], the energy of YSR states ($E_{\text{YSR}}$) with respect to the Fermi level ($E_{\text{F}}$) depends on the exchange coupling $J$ and the impurity spin $S_{\text{imp}}$:

$$E_{\text{YSR}} = \Delta \frac{1-a^2}{1+a^2} \quad \text{with} \quad a = JS_{\text{imp}}\pi\rho_s. \tag{1}$$

Here, $\Delta$ is the superconducting gap size and $a$ parameterizes the impurity-substrate interaction via $J$, $S_{\text{imp}}$, and $\rho_s$, which is the density of states of the substrate at $E_{\text{F}}$ in the normal state. From Eq. (1), it follows that YSR states may be localized at $E_{\text{F}}$, in which case they would be indistinguishable from a MZM in density of states measurements. This implies that a zero-bias peak (ZBP) in a density of states measurement is no definitive proof of MZMs (or TSC). Other arguments are thus required to label a ZBP as a MZM. For example, plateaus at quantized conductance values in transport experiments would support a MZM claim [29, 30].

Recently, the existence of a ZBP at the edges of $CrBr_3$ islands, grown on $NbSe_2$, has been interpreted as a 1D MZM [31]. It is reasoned that the magnetic insulator $CrBr_3$ interacts with the $s$-wave superconductivity of the $NbSe_2$ substrate, leading to YSR bands that are in a topological phase, thanks to a moiré potential [32]. The ZBP observation is supported by an extensive theoretical model, which can also explain the observed interruptions of the 1D MZM. Given the impact of this finding, we set out to confirm these observations and improve the understanding of this exciting material platform. Here, we report independent STM measurements of $CrBr_3/NbSe_2$ heterostructures, grown by MBE. In contrast to refs. [31, 32], we find no YSR bands in the interior of the $CrBr_3$ islands, and no 1D ZBP at the edges. Instead, we find edge-localized YSR states, whose energies are spread across the superconducting gap (including very close to zero energy). The YSR states disperse along small edge segments, and the YSR state energy can be tuned by interaction with the STM tip. Our findings are in excellent agreement with the recent report of Li et al. [33], which was published shortly after our experiments had been performed.

## 2 Methods

### Sample preparation

The growth of $CrBr_3$ thin films was performed in an ultra-high vacuum chamber with a base pressure of ca. $5 \times 10^{-10}$ mbar. A $NbSe_2$ single crystal (HQGraphene), glued to a molybdenum sample holder with conductive epoxy (EpoTek H20E), was used as the substrate. The $NbSe_2$ surface was prepared by cleaving, using scotch tape, in the fast entry lock (pressure lower than $4 \times 10^{-8}$ mbar). Prior to cleaving, the $NbSe_2$ sample was degassed up to 300 °C. $CrBr_3$ was evaporated from an e-beam evaporator (EFM3, Focus GmbH) containing a $CrBr_3$ single crystal (HQGraphene) in a quartz cup. A sub-monolayer coverage of crystalline $CrBr_3$ islands was obtained by evaporating for 6 min onto a freshly cleaved $NbSe_2$ crystal, which was heated to 230 °C. After growth, the sample was kept at the growth temperature for 15 min. After cooldown, the sample was transferred to the STM head, which is part of the same ultra-high vacuum system.

### STM/STS measurements

All scanning tunneling microscopy (STM) and scanning tunneling spectroscopy (STS) measurements were performed using a USM1300 (Unisoku Co. Ltd.), operated at $T = 360$ mK. A mechanically polished PtIr wire, conditioned on Au(111), was used as the STM tip. In STS experiments, dI/dV signals were recorded by lock-in detection, using a modulation frequency of either 771 Hz or 971 Hz. Modulation amplitudes $V_{ac}$ are given in the description of each experiment. STM and STS data was organized using SPMImageTycoon [34]; STM images were analyzed using Gwyddion [35].

### Shot noise measurements

We have measured the shot noise generated by the tunneling current at MHz frequency using a recently developed amplification circuit [36]. In essence, the circuit consists of a bias tee, a superconducting LC resonator ($f_0 = 2.8$ MHz, $Q \approx 1800$), and a high-electron-mobility transistor (HEMT). The bias tee is used to separate low frequency signals (STM feedback, dI/dV signal) from high frequency signals, whereas the LC resonator converts current noise from the junction into voltage noise at the gate of the HEMT. Lastly, the HEMT is used to match the high impedance circuit to the $50\,\Omega$ impedance of a transmission line. All these components are located in the vicinity of the 1 K pot, such that their temperature is expected to be $< 2$ K during the measurements at 360 mK ($T_{1K} \sim 1.6$ K). At room temperature, the noise signal is amplified by a 40 dB current amplifier (Femto HSA-X-1-40), before it is measured by a digital spectrum analyzer (MFLI, Zurich Instruments AG).

In a shot noise spectroscopy experiment, the magnitude of shot noise is measured at a set of bias voltages. In this report, shot noise spectroscopy was performed with the STM feedback configured to maintain a constant junction resistance $R_J = V_b/I$. We note that this implies a non-constant transparency of the tunnel junction, in the case of a non-constant density of states (such as a SC gap). To find the shot noise magnitude $S_I$, the power spectral density $S_V(\omega)$ is measured in a 100 kHz bandwidth around the resonator's center frequency, using lock-in detection (zoom FFT). For each bias point, the current noise is extracted from a fit of $S_V(\omega)$ using a simplified model of the circuit:

$$S_V(\omega) = S_{tot} \times |Z_{tot}(\omega)|^2 + S_V^0, \tag{2}$$

where $Z_{tot}(\omega)$ is the complex impedance of the resonator circuit, in parallel with the junction resistance $R_J$, and $S_V^0$ is the background (input) voltage noise level. The resonator circuit is

approximated by two parallel, identical RLC resonators ($R \sim 1\,\Omega$, $L = 120\,\mu\text{H}$, $C \sim 26\,\text{pF}$), separated by a coupling capacitor ($C_c = 100\,\text{pF}$). The fit parameters are $S_{\text{tot}}$, $S_V^0$, $R$ and $C$. $S_{\text{tot}} = G^2\left(S_I + S_I^0\right)$ is the total current noise, with $G$ being the gain of the amplification chain and $S_I^0$ being the background current noise level. The effective charge $Q_{eff}$ is extracted from the obtained $S_{\text{tot}}(V_b)$ values using

$$S_{\text{tot}} = G^2\left[2Q_{\text{eff}}|I| \times \coth\left(\frac{Q_{\text{eff}}V_b}{2k_B T}\right) + S_I^0\right], \tag{3}$$

where the Schottky formula for shot noise is recognized as $2Q_{\text{eff}}|I|$, and the coth-term accounts for thermal equilibrium noise of the tunnel junction [37]. Equation (3) is fitted to $S_{\text{tot}}(V_b)$ for bias points where the effective charge $Q_{\text{eff}}$ is known to be 1e, i.e., outside the SC gap. We find a gain $G$ of 0.176 (excluding room temperature amplification), whereas $S_I^0$ is ca. $50\,\text{fA}^2/\text{Hz}$. In our analysis, the thermal noise $S_I^{\text{th}} = 4k_B T/Z_{\text{tot}}$ is included in $S_I^0$ and not taken into account in the resonance circuit fit. By measuring each $S_V(\omega)$ for ca. 10 min (4000 FFT repetitions), we can determine $S_I$ to within $0.15\,\text{fA}^2/\text{Hz}$ (95 % confidence interval), independent of current/bias. This uncertainty is taken into account in the calculation of $Q_{\text{eff}}$, for which the confidence interval becomes larger for smaller currents, because of a lower signal-to-noise ratio.

# 3 Results and Discussion

## 3.1 Morphology & Structure

Figure 1a shows the typical morphology of the as-grown $CrBr_3/NbSe_2$ heterostructure. $CrBr_3$ has formed monolayer islands of triangular shape, which vary in size from 10 nm to 60 nm. The islands have an apparent height of $\sim 5.5\,\text{Å}$ at $V_b = 1.5\,\text{V}$ ($5.0\,\text{Å}$ at $V_b = 0.9\,\text{V}$), indicative of their monolayer thickness. In addition to the $CrBr_3$ islands, the evaporation process has introduced some organic contaminants, visible as elongated structures (annotated in blue). Small patches of $CrBr_2$ are also observed (annotated in red), always attached to $CrBr_3$ islands and indicative of slight $CrBr_3$ degradation [38].

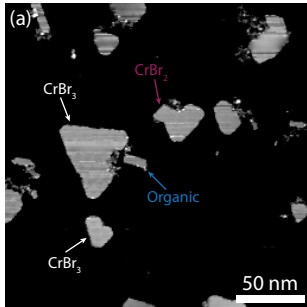 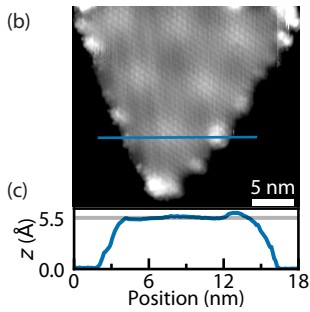 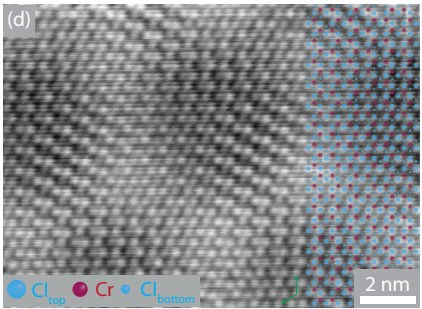

Figure 1: Structure of as-grown $CrBr_3/NbSe_2$ heterostructures at different length scales. (a) Morphology of the sample, showing that the $CrBr_3$ islands of different sizes coexist with small patches of $CrBr_2$ and some organic residue. Set point: 0.9 V, 50 pA. (b) Image of a single $CrBr_3$ island, on which both the atomic and moiré unit cells can be recognized. Set point: 1.5 V, 50 pA. (c) Line profile measured along the blue line shown in (b). The apparent height of the $CrBr_3$ island is 5.5 Å. (d) Image of the $CrBr_3$ lattice with atomic resolution: individual Br atoms are visible, as well as the moiré pattern. An atomic model of the $CrBr_3$ lattice is overlaid. Set point: 0.93 V, 50 pA.

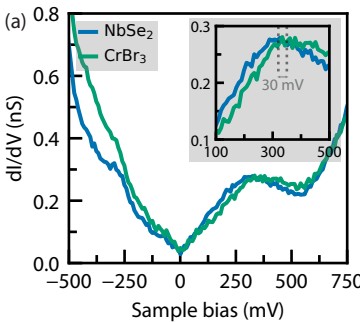
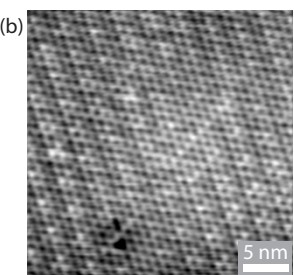
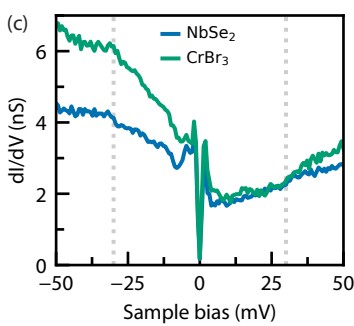

Figure 2: Characterization of the electronic structure of $CrBr_3$/$NbSe_2$. (a) dI/dV spectrum probing the charge transfer from $NbSe_2$ to $CrBr_3$. The Nb d-band at 300 mV shifts by ca. 30 mV, indicative of slight electron doping of $CrBr_3$. Set point: 1 V, 200 pA, $V_{ac}$ = 9 mV. (b) STM image showing the CDW of $NbSe_2$. Set point: 200 mV, 100 pA. (c) dI/dV spectrum probing the CDW of $NbSe_2$ beneath a $CrBr_3$ island. Set point: 100 mV, 300 pA, $V_{ac}$ = 0.65 mV.

On the $CrBr_3$ islands, a moiré pattern is visible, resulting from the lattice mismatch between $CrBr_3$ and $NbSe_2$ (see Fig. 1b). The moiré pattern has a periodicity of 6.1 nm ± 0.1 nm, which is similar to what has been reported previously [31, 33]. Furthermore, Fig. 1b shows that the $CrBr_3$ island is crystalline in the interior, but that the edges are not atomically straight – also in line with the previous reports. In an STM topograph with atomic contrast, shown in Fig. 1c, the moiré lattice can be observed together with the characteristic atomic pattern of $CrBr_3$. The moiré pattern is not exactly aligned with the rows of Br atoms, showing that the twist angle is not exactly 30° (as suggested in previous reports), but more close to 32°. Additionally, we observed that $CrBr_3$ islands on adjacent $NbSe_2$ planes, which have mirrored sublattice symmetries, have a mirrored orientation (see Fig. A1). This indicates that the twist angle is (in part) determined by the atomic interaction between the substrate and the epilayer.

## 3.2 Electronic Characterization

We have investigated how the magnetic insulator $CrBr_3$ affects the electronic structure of $NbSe_2$ via STS experiments inside the band gap of $CrBr_3$. By measuring inside the band gap, STS experiments can probe the $NbSe_2$ density of states (DOS) beneath the $CrBr_3$ islands. To study charge transfer effects, we recorded STS data in the energy range from $V_b = -0.5$ V to $V_b = 0.75$ V. By comparing data acquired on the $CrBr_3$ islands to data recorded on $NbSe_2$, we have found that the $CrBr_3$ induces a shift of ca. 30 mV of the Nb d-band at $V_b \sim 300$ mV. This shift has been interpreted as a sign of electron doping from $NbSe_2$ to $CrBr_3$ and is, given the uncertainty, similar in magnitude to what has been reported by refs. [31] and [33][*].

In the smaller bias range from $V_b = -100$ mV to $V_b = 100$ mV, we have used STS experiments to probe the charge density wave (CDW) beneath the $CrBr_3$ islands. Away from the islands, the CDW is readily visible in STM topographs such as Fig. 2b. In STS, the CDW is manifested as kinks at ca. ±35 mV [39]. As shown in Fig. 2c, we indeed observe a change of slope around ±30 mV, both on $NbSe_2$ (blue curve) and on $CrBr_3$ (green curve).

## 3.3 SC Gap Characterization

To investigate the effect of $CrBr_3$ on the superconductivity of $NbSe_2$, we performed STS experiments in a small bias window around $E_F$. First, we compared the SC gap measured on

---

[*]Ref. [31] reports a shift of +80 mV based on a contour plot, but from the line plot a value of +30 mV seems more appropriate.

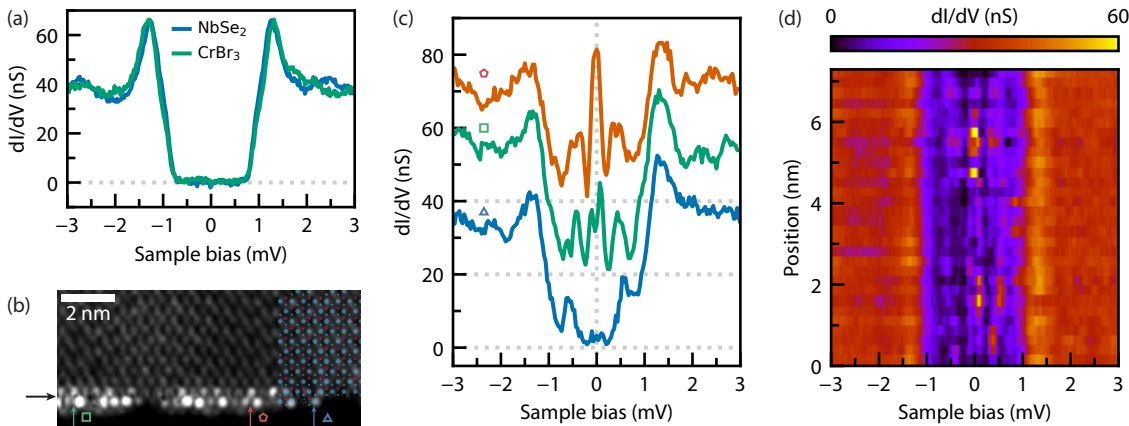

Figure 3: (a) SC gap of NbSe$_2$ (green) and CrBr$_3$ (blue). A hard SC gap is observed on both locations. (b) Constant height image of the edge segment probed by the STS experiments shown in (c), (d). Set point: 5 mV, 50 pA. (c) Selected spectra from the line spectrum shown in (d), displaying the variety of in-gap profile found on a single edge segment. Locations of the spectra are indicated in (b). Spectra are offset vertically by 20 nS for clarity. (d) Contour plot of dI/dV spectra acquired along a line on the edge shown in (b). The location of the line spectrum is indicated by the black horizontal arrow in (b). Set points for (a), (c) and (d): 5 mV, 150 pA, $V_{ac} = 33\,\mu$V.

the CrBr$_3$ islands to measurements on bare NbSe$_2$. We find that the SC gaps measured on both locations are nearly identical, as shown in Fig. 3a. In contrast to ref. [31], but in line with ref. [33], no YSR bands are found in the interior of the CrBr$_3$ islands. As a more detailed analysis, we have fitted the data of several NbSe$_2$ and CrBr$_3$ SC gaps using the McMillan model [40,41], but we do not observe a trend in the fit parameters indicative of YSR bands (see Appendix A.2). An investigation of the superconductivity in the interior of the CrBr$_3$ islands by shot noise spectroscopy is presented below.

Next, we focused on the edges of the CrBr$_3$ islands. By recording constant height maps, we have mapped the atomic structure of an edge (see Fig. 3b). In the interior of the island, the Br sublattice can be observed, from which we can extract the orientation of the CrBr$_3$ lattice (see the atomic model overlay in Fig. 3b). From the orientation, it follows that the edge corresponds to the armchair edge of the honeycomb sublattice formed by the Cr atoms. Even in close vicinity of the edge, the CrBr$_3$ crystallinity is pristine; only in the last unit cell ($\sim 6$ Å), the atomic structure is more disordered, and in some parts, missing atoms can be identified. However, in other edge segments, the CrBr$_3$ lattice appears intact, except for the unavoidable termination of the periodicity. In such parts, the outermost Br atoms are only faintly visible. The low intensity of these Br atoms suggests that they are bent downwards (towards the substrate), although we cannot rule out that this effect stems from the electronic structure instead. Either way, the presence of these Br atoms implies that the Cr atoms at the edge are sixfold coordinated, as they would be in the interior of the CrBr$_3$ island.

The constant-height current map (Fig. 3b) shows clear intensity variations along the edge. To explain this observation, we turn to the spectroscopy data acquired on this edge. dI/dV spectra acquired on positions that have different intensities in Fig. 3b exhibit very different in-gap features. The in-gap features come in pairs, which appear symmetrically around $E_F$, but are asymmetric in intensity. Based on the energy symmetry, we interpret the peak pairs to be YSR states. We note that these YSR states most likely do not result from undercoordinated Cr atoms at the edge, as the atomic structure is pristine (as discussed above). Instead, we expect the YSR states to arise from a Cr 3$d$ DOS at $E_F$ that is increased at the edge, as we

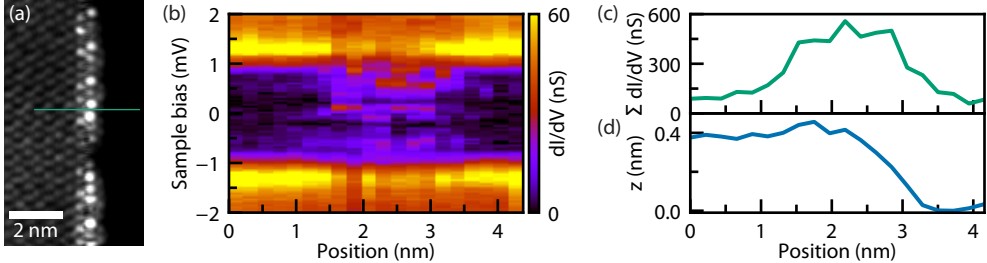

Figure 4: (a) Constant height map of a CrBr$_3$ island edge. The green line indicates the position of the line spectrum, which is shown in (b). Set point: 5 mV, 50 pA. (b) Line spectrum measured across a CrBr$_3$ edge. In-gap states are observed in a 2 nm wide region close to the edge. Set point: 5 mV, 150 pA, $V_{ac} = 33\,\mu$V (c), (d) Summed in-gap dI/dV signal and tip height of the line spectrum shown in (b).

have reported for CrCl$_3$/NbSe$_2$ [42]. The increased DOS strengthens the exchange interaction between the Cr atoms and the NbSe$_2$ substrate, and can explain the edge-localization of the YSR states.

Depending on position, the main YSR states can be located close to the coherence peaks (bottom curve), close to $E_F$ (middle curve), or precisely at $E_F$ (upper curve). In the latter case, the YSR states are so close in energy that only a single peak is observed at $E_F$. This happens despite the fact that in-gap states observed here have a smaller full-width at half maximum than those reported in ref. [31]. Not only the energy, but also the intensity of the YSR states can vary greatly, independent on the YSR state energy. To characterize the edge-localized YSR states more systematically, we have recorded a line spectrum along the edge (see the contour plot in Fig. 3d). The line spectrum shows that the YSR state energies vary smoothly with position, indicative of coupling between the individual YSR states. The coupling also extends across the low-intensity segment at ca. 3 nm from the left. The fact that the peak positions vary spatially, and are not pinned to zero energy, rules out the presence of topological superconductivity in our experiments. Instead, the YSR states are more likely to originate from the (magnetic) Cr atoms at the edge, similar to what we have observed for CrCl$_3$/NbSe$_2$ [42].

The constant height map of Fig. 3b suggests that the YSR states are localized to region of width $\sim$ 1 nm from the edge. To confirm this edge-localization, we performed dI/dV spectroscopy on a line perpendicular to the edge. Also in this line spectrum, the in-gap states appear within a small region close to the edge (see Fig. 4b). Compared to the high-intensity region in the constant height map of Fig. 4a, the YSR states in the line spectrum appear slightly more extended to the outside of the island. We attribute this to the fact that the feedback is enabled between each position in the line spectrum.

### 3.4 Further Characterization of YSR edge states

YSR states are sensitive to the chemical environment of the magnetic species. For example, it has been shown that the energy of YSR states is affected by the CDW of NbSe$_2$ [43] and by the moiré pattern of self-assembled molecular monolayers [44]. Here, we observed the YSR states at the CrBr$_3$ edges to be highly sensitive to the surrounding environment. In the CrBr$_3$ edge (line spectrum) shown in Fig. 3, the edge is well-ordered, resulting in an in-gap profile that changes smoothly along the edge. However, at an edge that is only slightly more disordered, we observed a variety of in-gap profiles, as shown in Fig. 5a, b. Spatially dispersive YSR states can still be recognized along small edge segments, but completely different in-gap profiles appear in adjacent segments. Most notably, the in-gap profile contains YSR states close to zero energy in the left part of the edge, whereas YSR states are only found closer to the coherence

peaks in the right part.

In addition to spatial variations, the YSR state energy could also be altered by interaction with the STM tip. By recording the SC gap at varying tip-sample distances (i.e., different tunneling transmissivity), the YSR state could be tracked through the quantum phase transition [28], as shown in Fig. 5c. At the largest tip-sample distance (lowest transmissivity, bottom of Fig. 5c), the most intense YSR states are found at $\pm 0.2$ meV. Upon decreasing the tip-sample distance (increasing transmissivity), these states shift towards $E_F$, up until a normal state conductance of $\sim 0.3\,\mu$S, after which the states split again. By changing the tip-sample distance, we tune the atomic forces between tip, substrate and epilayer. Presumably, the approaching tip pushes down the $CrBr_3$, thereby increasing the exchange coupling $J$ between the Cr atom(s) and the superconducting substrate [45–47]. The exchange coupling $J$ directly affects the YSR state energy $E_{YSR}$ according to Eq. (1), which is visualized in Fig. 5d. In the weak coupling regime (small $J$, $a^2 < 1$), the ground state of the spin-superconductor system is a free-spin state. In this regime, the in-gap states observed at positive (negative) energy in STS experiments correspond to the excitation to the screened-spin state via the electron-like (hole-like) part of the YSR state. When the exchange coupling is increased beyond a certain magnitude $J_{crit}$, the system goes through a quantum phase transition, after which the screened-spin state becomes the ground state. Beyond the phase transition, the electron- and hole-like components of the YSR state have switched sides, i.e., the electron-like part is found at negative energy, and the hole-like part at positive energy. In the STS data of Fig. 5c, this side-switching is visible by the reversal of the in-gap states' relative spectral weight. At the largest tip-sample distance, the in-gap state at negative energy is most intense, while the most intense state is found at positive energy for the smallest tip-sample distances. As may be clear from Fig. 5d, YSR states are located at $E_F$ when $J$ is tuned to be at the quantum phase transition ($J = J_{crit}$). In the current experiment, however, the steps in tip-sample distance were too coarse to observe this point of the phase diagram. Nevertheless, this experiment shows the tunability of the YSR states found at the edges of $CrBr_3$ islands.

## 3.5 Further Characterization of SC gap on $CrBr_3$ by Shot Noise Spectroscopy

A main difference between our data and that of ref. [31] is the absence of YSR bands in the SC gap, as probed by STS experiments inside the $CrBr_3$ islands (see Fig. 3a). To verify that the YSR bands are truly absent (and not just very small), we characterized the SC gap on $CrBr_3$ by means of regular STS experiments at varying tip-sample distance (Fig. 6a) and by shot noise spectroscopy (Fig. 6b, c). For the STS experiments, the current set point (at 5 mV) was increased from 300 pA for the largest tip-sample distance to 5 nA for the smallest tip-sample distance. In the resulting STS data, shown in Fig. 6a, no sign of YSR bands is observed at any tip-sample distance. Even at the smallest tip-sample distance, the SC gap is fully developed. At negative bias, a small feature can be observed inside the SC gap. We interpreted this feature as the wavefunction tail of one of the nearby ($\sim 6$ nm) edge-localized YSR states. The extension of YSR wavefunctions over such distances is possible in $NbSe_2$, thanks to the 2D character of its superconductivity [48].

In addition to these STS experiments, we have performed shot noise spectroscopy to check for YSR bands on $CrBr_3$. In shot noise spectroscopy, the magnitude of shot noise, measured as a function of the bias voltage, is used to determine the effective charge of the tunneling (quasi)particles. For a fully-developed superconducting gap, the effective charge changes from $Q_{eff} = 1$e at bias voltages outside the gap to $Q_{eff} = 2$e inside the gap, corresponding to a change in the tunneling mechanism from quasiparticle tunneling to Andreev reflections [49–52]. In tunnel junction shot noise experiments, the effective charge $Q_{eff}$ at in-gap biases is very sensitive to any 1e-tunneling pathways available, such as quasiparticle poisoning by temperature or (residual) magnetic fields. In constant conductance shot noise spectroscopy experiments,

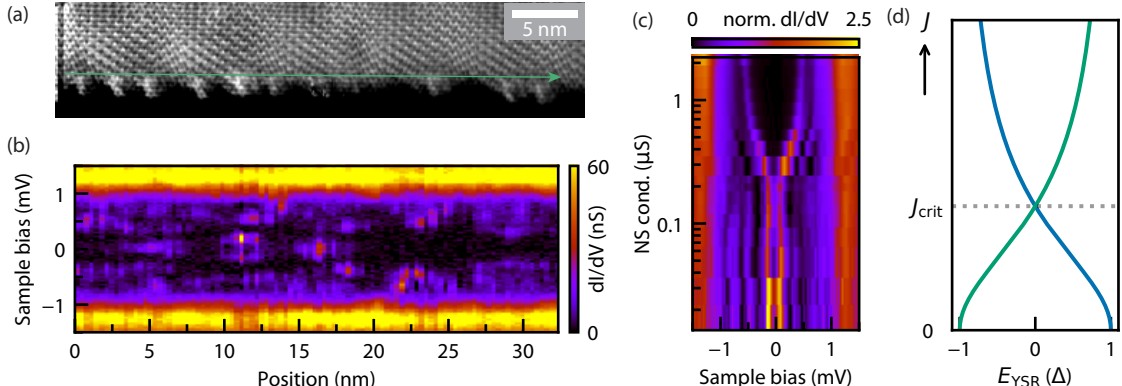

Figure 5: (a) Constant height STM image of an island edge that is longer but more disordered than the edge shown in Fig. 3b. Set point: 5 mV, 50 pA. (b) Contour plot of dI/dV spectra acquired along the green line shown in (a). In-gap states disperse along small edge segments only. Set point: 5 mV, 150 pA, $V_{ac} = 40 \mu V$. (c) Contour plot of dI/dV spectra of a YSR state, acquired at different tip-sample distances. By approaching the sample, the YSR state energy is tuned through the quantum phase transition. The normal-state (NS) conductance is defined as the average conductance signal in the bias range $|V_b| = 3$ mV to 5 mV. To calculate the normalized conductance, the differential conductance is divided by the NS conductance. Set points: 5 mV, 70 pA to 9 nA, $V_{ac} = 30 \mu V$. (d) Position of the YSR state energy as a function of the exchange interaction $J$, according to Eq. (1). The quantum phase transition occurs at $J = J_{crit}$.

as performed here, a $Q_{eff} = 1e$ contribution of only 1 % to the total tunnel current already reduces the effective charge from $Q_{eff} = 2e$ to $Q_{eff} \approx 1.1e$ [53, 54]. Here, we use this sensitivity to 1e-tunneling pathways to search for YSR bands that may be found in $CrBr_3/NbSe_2$. In typical STM experiments, the excitation of a YSR state is a single-electron process, which implies that tunneling events probing this excitation have an effective charge $Q_{eff} = 1e$ [55]. Hence, we expect to find an effective charge $Q_{eff}$ well below 2e on $CrBr_3/NbSe_2$ if any YSR bands are present [56]. Instead, we find an effective charge of $Q_{eff} = 1.7e \pm 0.1e$ (see Fig. 6b, c). Following ref. [53], this indicates that the contribution of 1e tunneling events is smaller than 0.1 %. The data in Fig. 6 confirm what is suggested by Fig. 3: no YSR bands exist in the $CrBr_3/NbSe_2$ heterostructure. Thus, the superconductivity is not affected by magnetic effects.

## 4 Conclusion

Two recent scanning tunneling microscopy studies have reported contrasting findings on the existence of topological superconductivity in $CrBr_3/NbSe_2$ heterostructures [31, 33]. Here, we have further characterized the $CrBr_3/NbSe_2$ system by STM, STS and shot noise spectroscopy, and found no sign of topological superconductivity (in agreement with ref. [33]). The most notable difference with respect to the earlier work reporting topological superconductivity (ref. [31]) is the observation of edge-localized YSR states, instead of a peak pinned at zero energy. Furthermore, in the interior of the $CrBr_3$ islands, we observe a clean SC gap, essentially without quasiparticles (< 0.1 %, as determined by shot noise spectroscopy). The spatial profile of the YSR states is reminiscent of the one we have recently observed in the related heterostructure $CrCl_3/NbSe_2$ [42]. In analogy to the case of $CrCl_3/NbSe_2$, the edge-localized YSR states presumably reflect an increase of the $Cr-NbSe_2$ exchange interaction, which occurs

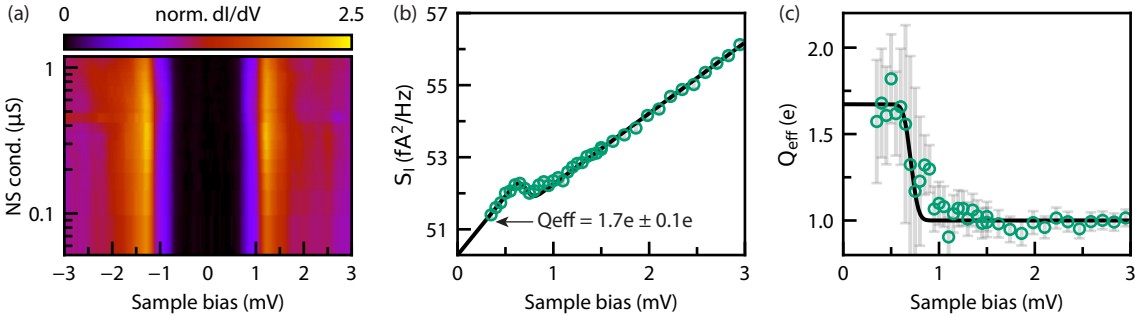

Figure 6: (a) Contour plot of dI/dV spectra of a YSR state, acquired at different tip-sample distances. The SC gap is not affected by the tip-sample distance. The NS conductance is calculated as in Fig. 5b. Set points: 5 mV, 300 pA - 5 nA, $V_{ac} = 30\,\mu V$. (b), (c) Current noise $S_I$ (b) and effective charge $Q_{eff}$ (c) when tunneling into $CrBr_3$, as determined by shot noise spectroscopy. A constant junction resistance $R_J = 5\,M\Omega$ was maintained during the experiment. Inside the SC gap, an effective charge $Q_{eff} = 1.7e \pm 0.1e$ is found. Error bars indicate 95 % confidence intervals.

even at perfect, unreconstructed $CrBr_3$ edges.

At present, it is unclear what causes the absence (here, ref. [33]) or presence (ref. [31]) of in-gap states in the interior of $CrBr_3$ islands on $NbSe_2$. The difference may be caused by a different substrate-epilayer interaction or, possibly, the presence of defects/impurities. A systematic study into if/how the properties of $CrBr_3/NbSe_2$ depend on the growth conditions may shed light on the origin of the different experimental observations. Additionally, more insight into topological superconductivity in $CrBr_3/NbSe_2$ may be obtained by the investigation of other magnetic 2D materials proximitized to *s*-wave superconductors.

# Acknowledgments

The authors thank P. Liljeroth and S. Kezilebieke for insightful discussions.

**Author contributions**   I.S. conceived the project. J.P.C. performed the experiments and analyzed the data. D.V. and I.S. supervised the experiments. All authors discussed the results and contributed to writing of the manuscript.

**Conflicting interests**   The authors report no conflicting interests.

**Funding information**   I.S. thankfully acknowledges funding by the European Research Council (Horizon 2020 "FRACTAL", 865570). D. V. and I. S. acknowledge the research program "Materials for the Quantum Age" (QuMat) for financial support. This program (Registration Number 024.005.006) is part of the Gravitation program financed by the Dutch Ministry of Education, Culture and Science (OCW).

**Data availability**   The data used to produce the figures in this manuscript have been published and can be downloaded from [57].

# A Extended Data

## A.1 Mirrored CrBr$_3$ orientation on adjacent NbSe$_2$ planes

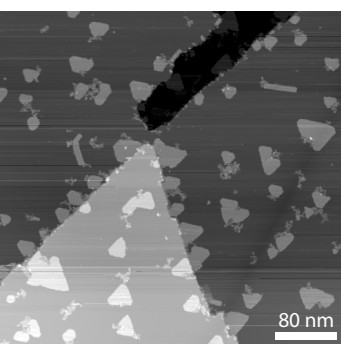

Figure A1: STM image of a location on the sample where different NbSe$_2$ terraces are observed. CrBr$_3$ islands on adjacent NbSe$_2$ terraces have opposite orientations. Set point: 0.9 V, 50 pA.

## A.2 McMillan fitting of SC gap

We have fitted the SC gap, as measured on different position on the sample and with different nanoscopic tips, using the McMillan model for two-band superconductivity. Following ref. [41], the dI/dV signal $G_{\text{tot}}$ was fitted to:

$$G_{\text{tot}}(E) = \sum_{i=1,2} G_{\text{i}} \text{Re} \left[ \frac{|E|}{\sqrt{E^2 - \Delta_i^2(E)}} \right], \tag{4}$$

where $G_{\text{i}}$ is the weighing factor for the two bands $i = \{1, 2\}$, and $\Delta_i(E)$ is the energy-dependent gap magnitude, which is calculated self-consistently from:

$$\Delta_i(E) = \frac{\Delta_i^0 + a_{ij} \Delta_j(E) / \sqrt{\Delta_j^2(E) - E^2}}{1 + a_{ij} / \sqrt{\Delta_j^2(E) - E^2}}. \tag{5}$$

In Eq. (5), $\Delta_i^0$ is the intrinsic gap in band $i$ and $a_{ij}$ quantifies the coupling between the two bands $i$ and $j$ ($i \neq j$). To include (thermal) broadening effects, Eq. (4) was convoluted by the derivative of the Fermi-Dirac equation with an effective temperature $T_{\text{eff}}$. As shown in Fig. A2, we obtain good agreement with the experimental data. We find an effective temperature of ca. 500 mK for all datasets. From the parameters of all the fits, which are shown in Table A1, we do not observe a trend that suggests a filling of the SC gap on CrBr$_3$ positions.

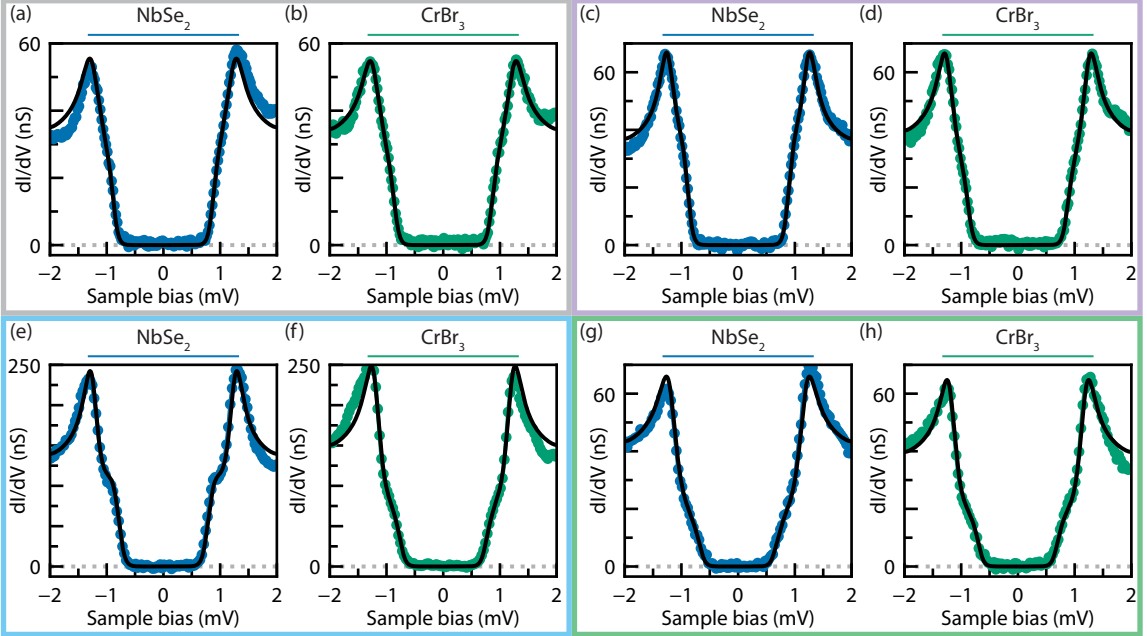

Figure A2: STS data of the SC gap on NbSe$_2$ and CrBr$_3$ positions, acquired on four different, macroscopic locations on the sample and with different nanoscopic tip. Fits to the McMillan model (Eqs. (4) and (5)) are shown as black lines. Pairs of measurements on NbSe$_2$ and CrBr$_3$, as indicated by the color coding, are performed consecutively, and with the same tip and the same measurement parameters. STM feedback was disabled at 5 mV, at a current set point of 150 pA [(a)-(d), (g), (h)] or 500 pA [(e), (f)]. Lock-in amplitude $V_{ac}$ was either 33 μV [(a)-(d)] or 30 μV [(e)-(h)].

| Panel | Position | $T_{eff}$ (K) | $\Delta_1^0$ (meV) | $G_1$ (nS) | $a_{12}$ (meV) | $\Delta_2^0$ (meV) | $G_2$ (nS) | $a_{21}$ (meV) |
|---|---|---|---|---|---|---|---|---|
| (a) | NbSe$_2$ | 0.49 | 1.32 | 26 | 0.54 | 0.0 | 4.5 | 3.37 |
| (b) | CrBr$_3$ | 0.50 | 1.31 | 26 | 0.53 | 0.0 | 4.2 | 3.55 |
| (c) | NbSe$_2$ | 0.50 | 1.26 | 23 | 0.28 | 0.0 | 9.4 | 3.32 |
| (d) | CrBr$_3$ | 0.50 | 1.29 | 30 | 0.39 | 0.0 | 4.2 | 3.36 |
| (e) | NbSe$_2$ | 0.56 | 1.26 | 75 | 0.17 | 0.0 | 32.7 | 2.20 |
| (f) | CrBr$_3$ | 0.53 | 1.24 | 104 | 0.27 | 0.1 | 7.7 | 2.03 |
| (g) | NbSe$_2$ | 0.52 | 1.23 | 32 | 0.35 | 0.0 | 0.0 | 1.96 |
| (h) | CrBr$_3$ | 0.53 | 1.22 | 28 | 0.27 | 0.0 | 0.3 | 1.97 |

Table A1: Optimized parameters obtained by fitting the McMillan model to dI/dV spectra of the SC gap on NbSe$_2$ and CrBr$_3$ positions. The first column ('Panel') refers to the panels in Fig. A2, where the data and the fits are shown. The parameters are introduced in Eqs. (4) and (5).

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
