# Peer review of "Non-Topological Edge-Localized Yu-Shiba-Rusinov States in CrBr$_3$/NbSe$_2$ Heterostructures"

_SciPost Physics_

## Round 1 · Referee Report · Anonymous (Referee 1) · 2025-2-22

Strengths

This paper discusses the nature of edge states in CrBr3/NbSe2 heterostructures. Such hybrid ferromagnet-superconductor heterostructures have attracted extensive attention as they potentially host topological superconductivity characterized by the existence of Majorana Zero Mode (MZM) localized at the edges of the structure. While the existence of a Zero Bias Peak (ZBP) in the differential conductance probed by tunneling spectroscopy at the edges of CrBr3 islands grown on NbSe2 has been interpreted as the signature of an MZM by some authors, this interpretation has been discarded by others due to the continuous evolution of such a ZBP from a single peak at zero energy to two peaks (symmetric with respect to zero energy) as the tunneling transmittivity is varied - a behavior consistent with topologically trivial Yu-Shiba-Rusinov (YSR) states.
The authors report low temperature scanning tunneling microscopy and spectroscopy measurements in a clear and rigorous manner: - they measure the superconducting gap of NbSe2 and NbSe2 capped with a monolayer of CrBr3 - They present shot noise measurements that allow to determine the effective charge of the tunneling quasiparticles and thus the existence of YSR bands inside the island. The measured effective charge of 1.7e (close to 2e) indicates that superconductivity is not affected by magnetic effects - they show that the edge states are sensitive to disorder, which is unexpected for topologically protected edge states - they use the sample-tip distance as a tool to vary the exchange coupling which directly affects the energy of YSR states. The observed evolution of differential conductance peaks inside the superconducting gap is consistent with the one expected for YSR and inconsistent with the interpretation of ZBP as a signature of MZM

Their results are consistent with the egde states in CrBr3/NbSe2 being of topologically trivial nature and underline the necessity of using several probes when addressing the topological nature of edge states

Report

This paper meet the journal's acceptance criteria in term of novelty. In my opinion, it also has potential for multi-pronged follow-up work to investigate edge states in 2D topological superconductor candidates

Recommendation

Publish (meets expectations and criteria for this Journal)

---

## Round 1 · Referee Report · Anonymous (Referee 2) · 2025-3-1

Strengths

  1. The paper directly addresses a key debate in condensed matter physics, challenging previous claims of Majorana zero modes (MZMs) and offering a well-supported alternative explanation.
  2. The experimental data is of high quality and complementary to prior publications.
  3. The conclusions are sound and the paper well written.

Report

This study investigates the superconducting properties of CrBr₃/NbSe₂ heterostructures to clarify whether topological superconductivity emerges in these systems. Previous research by Kezilebieke et al. (Nature 588(7838), 424 (2020), Ref. 31) had suggested the presence of interfacial topological superconductivity and one-dimensional Majorana zero modes (MZMs) at the edges of the CrBr₃ islands. Using low-temperature scanning tunneling microscopy (STM) and shot noise spectroscopy, the Cuperus and colleagues now find that the superconducting gap inside CrBr₃ remains unaffected by magnetism, contradicting prior claims. At the island edges, Yu-Shiba-Rusinov (YSR) states are observed at various in-gap positions, including near zero energy. However, these states disperse spatially and shift upon approaching with the STM tip. Hence, they do not form a robust zero-energy peak characteristic of MZMs. The results show that the observed in-gap states are conventional YSR states rather than topological signatures.
The authors propose that the Cr 3d bands at the edges of the CrBr₃ islands exhibit an increased density of states (DoS), which enhances the exchange interaction and drives the formation of YSR states. The absence of YSR bands within the CrBr₃ islands is confirmed by transmission-dependent dI/dV spectra and further reinforced by shot noise spectroscopy. The latter provides a highly sensitive validation, demonstrating that the magnetic insulator has minimal impact on the superconducting substrate.
These findings are in agreement with a recent study by Li et al. (Nat. Commun.15, 10121 (2024), Ref. 33), which also reported the absence of topological superconductivity in CrBr₃/NbSe₂. By reinforcing these conclusions with additional shot noise spectroscopy, this work challenges earlier claims of topological superconductivity and highlights the need for a more nuanced understanding of magnet-superconductor interactions.

Requested changes

Minor suggestions for improvement: 1. Given the importance of the atomic and electronic structure of the island edges, Fig. 3b would benefit from a larger presentation. 2. Is there any spectroscopic indication of the enhanced DoS of the Cr 3d states at the edges?

Recommendation

Publish (easily meets expectations and criteria for this Journal; among top 50%)

---

## Round 1 · Referee Report · Anonymous (Referee 3) · 2025-3-24

Strengths

1- The manuscript shows that bulk Yu-Shiba-Rusinov states in CrBr3/NbSe2 can be absent, complementing past experiments showing their presence 2- Associated with absent bulk Yu-Shiba-Rusinov states, fragile edge Yu-Shiba-Rusinov states are observed 2-The manuscript presents an experiment of a CrBr3/NbSe2 with clear signatures of a topological trivial state and trivial edge modes

Weaknesses

1- No weaknesses

Report

The authors study a NbSe2/CrBr3 heterostructure, focusing on the interplay between the superconducting state and magnetism. The coexistence of these two orders represents one of the strategies to engineer topological superconductivity. Such a potential state arises in the presence of topological Yu-Shiba-Rusinov bands in the bulk of the heterostructure, leading to topological zero modes in their experiment. The authors show that while ingap states appear at the edges, no bulk Yu-Shiba-Rusinov states are present. Such observation is incompatible with the existence of topological superconductivity, which leads to the conclusion that their heterostructure is topologically trivial. This is further supported by controlling the exchange coupling between the edge moments and the superconductor by means of the STM.

Interestingly, the phenomenology observed in the experiment is different from previous experiments in analogous heterostructures, that show the existence of bulk Yu-Shiba-Rusinov states, and whose edge zero modes were rationalized as stemming from a topological superconducting state. The current experiment demonstrates that widely different phenomena can appear in a similar heterostructure, which in particular can be associated with a trivial superconducting state.

I find their results highly interesting, and their demonstration of the trivial state in their heterostructure is very robust from my perspective. I believe that the demonstration of the absence of bulk Yu-Shiba-Rusinov states in the same heterostructure where they were observed before is of great interest to the community, and in particular it highlights how similar heterostructures can feature widely different phenomena. Their manuscript is very well written and the conclusions are well justified. Given all the points above, I strongly recommend their manuscript for publication in Scipost Physics.

Besides my positive recommendation, I would like to ask the authors if they could comment on what could be the potential reason for the absence of bulk Yu-Shiba-Rusinov states in their experiment, in comparison with their presence in Ref. 31. From my perspective, one potential mechanism could be a different twisting angle between CrBr3 and NbSe2 in their experiment and Ref. 31, as the twisting angle has been shown to strongly impact exchange couplings between van der Waals materials (Phys. Rev. B 107, 035112 (2023)). Another potential mechanism could be a different sample strain between their experiment and Ref. 31. Of course, the authors would not need to make strong statements about this in the revised version. Nevertheless if for example, the twisting angle of their heterostructure were different than Ref. 31, it would be great that they note in their revised version that this could be a potential mechanism.

To summarize, I believe that the current manuscript provides an experiment of exceptional interest to the community, highlighting the widely different physics that CrBr3/NbSe2 may show. I thus strongly recommend their manuscript for publication.

Requested changes

1 - It can be nice adding a short discussion about the potential role of strain or different twist angle for the absence of bulk Yu-Shiba-Rusinov states

Recommendation

Publish (surpasses expectations and criteria for this Journal; among top 10%)

---

## Round 1 · Referee Report · Anonymous (Referee 4) · 2025-3-31

Strengths

The experimental data is of extremely high quality and the experiments were carefully performed. The addition of shot-noise based measurements is also a novelty that to date has not been extensively used in these types of measurements.

Weaknesses

(1) my main criticism is of the YSR treatment in this paper. It was confusing for me if the authors claim this is a quantum spin or classical spin system. They in the end use a spin ½ treatment to explain their data (e.g. Fig. 5), which is not appropriate in the case that there is a high spin. The data could also be understood in terms of having different types of isolated YSR states of a high-spin character. Experimentally they observe multiple peaks, and they cannot exclude the possibility that this is a high spin YSR state (as opposed to some band). In both cases, there is extensive theoretical and experimental literature, and I do not think they can claim to understand this in the simplest limit. For example, changing the exchange field, e.g. in Fig. 5b, may also be highly non-linear in the limit of high spin (see e.g. PRB, 105, 235406 (2022), and PRB, 103, 205424, (2021)). I can understand that the spin ½ picture is the easiest to interpret, but I believe that it is important considering they observe something in contrast to published reports, to also loosen this claim and explain the other theoretical interpretations of the data (e.g. claiming a QPT). (2) I missed a bit of a more in-depth discussion at the end of the paper to describe the differences between this work and the previous papers. I would recommend expanding the last paragraph in the paper. (3) there are some minor details that may need some changes, which I elaborate below. In general, I would consider this for publication.

Report

The paper from Cuperus et al is an experimental study using low temperature STM to explore the superconductivity and YSR states of the monolayer of CrBr3 grown on bulk NbSe2. The main observations of the paper are in juxtaposition with earlier STM studies of the same material, and contrasting observations. They show that they can synthesize the layer CrBr3 layer with high quality, along with other stoichiometries. In contrast to previous reports, where a zero-bias peak was observed at the edges of the island structures, they observe multiple in-gap states away from zero bias. Thus, they claim to not observe signatures of Majorana end modes in STM spectroscopy, and confer this with new shot-noise measurements.
I recommend publishing this paper after some revision. In general, the experimental quality of the paper is high. I found three things that require revision, but otherwise I would support publication in SciPost: (1) my main criticism is of the YSR treatment in this paper. It was confusing for me if the authors claim this is a quantum spin or classical spin system. They in the end use a spin ½ treatment to explain their data (e.g. Fig. 5), which is not appropriate in the case that there is a high spin. The data could also be understood in terms of having different types of isolated YSR states of a high-spin character. Experimentally they observe multiple peaks, and they cannot exclude the possibility that this is a high spin YSR state (as opposed to some band). In both cases, there is extensive theoretical and experimental literature, and I do not think they can claim to understand this in the simplest limit. For example, changing the exchange field, e.g. in Fig. 5b, may also be highly non-linear in the limit of high spin (see e.g. PRB, 105, 235406 (2022), and PRB, 103, 205424, (2021)). I can understand that the spin ½ picture is the easiest to interpret, but I believe that it is important considering they observe something in contrast to published reports, to also loosen this claim and explain the other theoretical interpretations of the data (e.g. claiming a QPT). (2) I missed a bit of a more in-depth discussion at the end of the paper to describe the differences between this work and the previous papers. I would recommend expanding the last paragraph in the paper. (3) there are some minor details that may need some changes, which I elaborate below. In general, I would consider this for publication.
Detailed points:
(1) I found a bit confusing: is the CrBr3 layer insulating? Do they have experimental evidence of this? For example: wouldn’t this suppress Andreev reflections (e.g. in their shot noise discussion)?
(2) Could the authors specify their experimental resolution and the tip type?
(3) Is hydrogen an important consideration, for example in comparing differences?
(4) In Fig. 1, what are the clusters on the edge of the island? Won’t this create disorder?
(5) Did the authors apply magnetic field to quench the superconducting state and see if there’s residual LDOS at the island edge, i.e. if it’s metallic? I found this confusing, as e.g. p7 the authors write that the “increased DOS strengthens the exchange interaction…” If it’s insulating, it isn’t clear why this would matter.
(6) Could the authors discuss other things that may change if the layer relaxes when the tip is approached? Hypothetically, there can be a different charge transfer, magnetic moment, or ultimately magnetic anisotropy: all matter if it is beyond spin 1/2 .

Requested changes

(see above)

Recommendation

Ask for minor revision

---

## Editorial Decision

resubmitted